# The Area and Number of Intraretinal Cystoid Spaces Predict the Visual Outcome after Ranibizumab Monotherapy in Diabetic Macular Edema

**DOI:** 10.3390/jcm9051391

**Published:** 2020-05-08

**Authors:** Norihiro Nagai, Misa Suzuki, Atsuro Uchida, Toshihide Kurihara, Norimitsu Ban, Sakiko Minami, Hajime Shinoda, Kazuo Tsubota, Yoko Ozawa

**Affiliations:** 1Laboratory of Retinal Cell Biology, Keio University School of Medicine, 35 Shinanomachi, Shinjuku-ku, Tokyo 160-8582, Japan; nagai@a5.keio.jp (N.N.); misayutakatomo@icloud.com (M.S.); 2Department of Ophthalmology, Keio University School of Medicine, 35 Shinanomachi, Shinjuku-ku, Tokyo 160-8582, Japan; uchidats@gmail.com (A.U.); kurihara@z8.keio.jp (T.K.); nban@keio.jp (N.B.); saki.love5@icloud.com (S.M.); shinoha@mac.com (H.S.); tsubota@z3.keio.jp (K.T.); 3Department of Ophthalmology, St. Luke’s International Hospital, 9-1 Akashi-cho, Chuo-ku, Tokyo 104-8560, Japan; 4St. Luke’s International University, 9-1 Akashi-cho, Chuo-ku, Tokyo 104-8560, Japan

**Keywords:** diabetic macular edema, optical coherence tomography, intraretinal cystoid space, predictive factor, biomarker

## Abstract

Visual outcomes in diabetic macular edema (DME) after anti-vascular endothelial growth factor therapy vary across individuals. We retrospectively reviewed the clinical records for 46 treatment-naive eyes of 46 patients with DME who underwent intravitreal ranibizumab (IVR) monotherapy with a pro re nata regimen for 12 months. Overall, mean best-corrected visual acuity (BCVA) improved. Multivariate analyses adjusted for age and baseline BCVA showed that the area ratio, compared with the retinal area, and the number of intraretinal cystoid spaces evaluated on OCT (optical coherence tomography) images at baseline positively correlated with LogMAR BCVA and the extents of ellipsoid zone and external limiting membrane disruption at 12 months, and negatively correlated with central retinal thickness at the time of edema resolution. Therefore, a high area ratio and large number of intraretinal cystoid spaces resulted in a disorganized outer retinal structure at 12 months, a thin and atrophic retina after edema resolution, and a worse visual outcome. The area ratio and number of intraretinal cystoid spaces on initial OCT images were predictors of the visual outcome after IVR therapy in DME irrespective of baseline age and BCVA. The factors were related to retinal neurodegenerative changes in DME and could help in obtaining proper informed consent before treatment.

## 1. Introduction

Diabetic retinopathy (DR) is a major complication of diabetes mellitus and a leading cause of visual impairment [1,2]. In contrast to proliferative DR, diabetic macular edema (DME) can occur at any severity of the DR stage, even in simple DR [3], and affects the macula to impair central visual acuity, color perception, and the quality of vision [4]. The administration of anti-vascular endothelial growth factor (VEGF) therapy, which suppresses the VEGF-induced breakdown of the blood–retinal barrier (BRB) and resulting extracellular fluid accumulation, improves the prognosis of patients with DME [5,6,7,8]. However, the treatment outcomes vary substantially among patients. DME is a chronic disease and requires repeated injections of anti-VEGF drugs. To obtain informed consent for treatment, the patient should have a proper understanding of the expected outcome. Thus, predictive factors that affect visual outcomes would be of value. 

Spectral domain optical coherence tomography (SD-OCT) is commonly used to evaluate DME before and during treatment. OCT images show the presence of edema and provide varied information regarding the condition of the retina. Previous reports have shown that DME can be divided into three categories characterized by the distribution of the extracellular fluid [9], i.e., sponge-like retinal swelling, cystoid macular edema (CME), and serous retinal detachment (SRD). The efficacy of DME treatment according to the initial OCT findings has been analyzed according to the initial categories [10,11]. However, the results have been controversial among reports [10,11], and an overlap of characteristics is sometimes observed, which complicates estimating the prognosis of particular patients [9]. 

Other predictive factors of treatment outcome involve initial best-corrected visual acuity (BCVA) [12,13], central retinal thickness (CRT) [14,15], and outer retinal layer thickness, which correspond to the inner and/or outer segment lengths of the photoreceptors [16] on OCT images. However, with the recent trend of early diagnosis and early treatment, treatment may be started before visual disturbance and/or photoreceptor morphological changes become severe [17]. Information derived from OCT images that represents retinal pathological conditions and could predict the outcome independently of BCVA would be of value and need to be studied further.

In this study, we evaluated initial OCT images to determine biomarkers that would be easily applicable in daily clinical practice to predict visual outcomes and to elucidate the underlying pathogenesis of the DME-affected neural condition of the retina, aiming at helping patients and clinicians to plan the treatment and explore the mechanisms of retinal neural damage due to DME.

## 2. Experimental Section

### 2.1. Participants

This was a retrospective study based on a detailed medical chart review. The study included 46 eyes of 46 patients with treatment-naïve DME that were treated with intravitreal ranibizumab (IVR) at the Medical Retina Division Clinic of the Department of Ophthalmology, Keio University Hospital (Tokyo, Japan) between April 2014 and October 2018. All patients attended the clinic for at least 12 months, during which the only treatment they received was IVR. 

### 2.2. Eye Examinations

All patients underwent BCVA measurement with the refraction test, a slit-lamp examination, and binocular indirect ophthalmoscopy after pupil dilation with 0.5% tropicamide. These examinations were performed at every follow-up visit. Follow-up was carried out every month, but when intra- and/or sub-retinal exudative changes were absent for more than 2 months, the interval was extended to up to every 2 months until recurrent exudative changes were observed. Exudative changes were mainly evaluated by OCT images, referring to the results of fundus examinations. The severity of DR was graded by using fundus photographs and referring to the Fluorescein Angiography (FA) at baseline. 

### 2.3. Fluorescein Angiography

Fluorescein Angiography (FA) was performed using a Topcon TRC50DX retinal camera (Topcon, Tokyo, Japan) to determine the grade of the DR. 

### 2.4. Optical Coherence Tomography

Horizontal sections of Optical Coherence Tomography (OCT) images passing through the fovea were obtained with a Heidelberg Spectralis OCT (Heidelberg Engineering, Dossenheim, Germany) instrument in OCT Section mode (30°, average of a maximum of 100 frames acquired using a retinal tracking system). Repeat mode was applied at the time of the follow-up visits to obtain OCT images at the exact same location. The images were used to evaluate the CRT, intraretinal cystoid space and subretinal fluid (SRF), and hyperreflective foci. The CRT was defined as the distance between the internal limiting membrane and the presumed retinal pigment epithelium (RPE) at the fovea. The following parameters were evaluated on horizontal OCT images within a 2000 μm diameter around the foveal center. The total area of intraretinal cystoid spaces and the retinal area were calculated using Image J (National Institutes of Health, Bethesda, MD, USA), and the area ratio of the intraretinal cystoid spaces to the retina was defined as the area ratio of the cystoid spaces. SRF was not involved either in the intraretinal cystoid space or retinal areas. The number of intraretinal cystoid spaces with a width larger than 200 μm was counted. The extents of the disorganization of the retinal inner layers (DRIL) [18,19,20] and ellipsoid zone (EZ) and external limiting membrane (ELM) disruptions were measured. If there were no DRIL, or disruptions of the EZ or ELM, the values were set to 0. Measurement was performed by referring to the scale bars in the OCT system and the software embedded in the SD-OCT system. The OCT imaging was performed at every follow-up visit. 

### 2.5. Anti-VEGF Therapy

Ranibizumab (0.5 mg, 0.05 mL, Genentech, Inc., South San Francisco, CA, USA) was injected intravitreally under sterile conditions via the pars plana at first injection. Re-injections were recommended if the OCT image and/or fundus examination showed evidence of any exudative changes in the macula, identified as macular edema or SRF at the time of the follow-up examinations, and performed under informed consent. 

### 2.6. Statistical Analyses

All results are expressed as the mean ± standard error. Commercially available software (SPSS, V. 25.0, IBM Corp, Armonk, NY, USA) was used for the statistical analyses. The demographic characteristics of the baseline and outcomes were compared using the repeated measure ANOVA, Pearson correlation coefficients, stepwise multiple linear regression models, and multiple regression analyses. Statistical significance was set at *p* < 0.05.

## 3. Results

### 3.1. Baseline Characteristics and OCT Findings

Forty-six eyes of 46 patients with DME (29 men, 17 women; mean age, 65.5 ± 1.6 years; range, 40 to 80 years) were included in this study. The baseline characteristics of the patients are shown in Table 1. Thirteen eyes (28.26%) had mild to moderate non-proliferative DR, 20 eyes (43.48%) had severe non-proliferative DR, and 13 eyes (28.26%) had proliferative DR. Twenty-six eyes (56.5%) had past histories of pan-retinal photocoagulation. 

The mean area ratio of the intraretinal cystoid space to the retinal area was 19.7% ± 1.9%, and the mean number of cystoid spaces was 3.7 ± 0.3; the area ratio was calculated using the data of absolute amounts of the intraretinal cystoid space (mean amount, 8356 ± 994 pixels^2^; range, 997–37,290 pixels^2^) and the retinal (mean amount, 39,895 ± 1442 pixels^2^; range, 2188–72,565 pixels^2^) areas. The mean CRT, extents of DRIL, and ELM and EZ disruptions were 418 ± 21 μm, 1056 ± 51 μm, 273 ± 80 μm, and 580 ± 95 μm, respectively. Sixteen eyes (34.8%) had hyperreflective foci, and nine eyes (19.6%) had SRF at baseline. All eyes (100%) exhibited CME, 37 eyes (80.4%) were categorized into the CME type with no SRD type, nine eyes (19.6%) were SRD type, and no eyes were sponge-like retinal swelling type. [9] There were no eyes with pre- or intra-retinal hemorrhage greater than 1/2 disc diameter at the foveal region.

### 3.2. Treatment Outcome of IVR Monotherapy for DME

The mean BCVA improved (0.287 ± 0.038 at baseline; 0.217 ± 0.044 at 12 months, *p* = 0.009, Figure 1a) and the mean CRT decreased (418 ± 21 μm at baseline; 276 ± 20 μm at 12 months, *p* < 0.001, Figure 1b) at 3 and 12 months after initial IVR treatment compared with baseline. 

The area ratio of the cystoid space (19.7% ± 1.9 % at baseline; 12.5% ± 1.7% at 12 months, *p* = 0.005, Figure 1c) and the number of cystoid spaces (3.7 ± 0.3 at baseline; 1.7 ± 0.2 at 12 months, *p* < 0.001, Figure 1d) also significantly decreased at 3 and 12 months compared with baseline. 

The mean extent of the DRIL (1056 ± 51 μm at baseline; 702 ± 69 μm at 12 months, *p* < 0.001, Figure 1e) and the mean extent of ELM (273 ± 80 μm at baseline; 117 ± 49 μm at 12 months, *p* = 0.038, Figure 1f) and EZ (581 ± 95 μm at baseline; 270 ± 79 μm at 12 months, *p* < 0.001, Figure 1g) disruptions decreased at 12 months compared with baseline. DRIL and EZ disruptions were already decreased, and ELM disruption showed a decrease at 3 months.

The mean number of injections was 3.9 ± 0.4 during the 12-month follow-up period.

### 3.3. Association between BCVA at 12 Months and the Baseline Findings

Univariate analyses revealed that the area ratio (*R* = 0.688, *p* < 0.001) (Table 2, Figure 2a) and number (*R* = 0.557, *p* < 0.001) (Table 2, Figure 2b) of the intraretinal cystoid spaces at baseline were correlated with BCVA at 12 months. The baseline BCVA (*R* = 0.830, *p* < 0.001), CRT (*R* = 0.593, *p* < 0.001), and extents of DRIL (*R* = 0.593, *p* < 0.001) and ELM disruption (*R* = 0.405, *p* = 0.005) were also associated with BCVA at 12 months (Table 2).

We further analyzed the correlations using stepwise multiple linear regression models (Table 2). After adjusting for age and baseline BCVA, BCVA at 12 months was positively correlated with the area ratio (95% confidence interval (CI), 0.006 to 0.013; *p* < 0.001), and number (95% CI, 0.008 to 0.059; *p* = 0.012) of the cystoid spaces at baseline. There were no significant correlations between BCVA at 12 months and other baseline findings.

Therefore, a high area ratio and large number of intraretinal cystoid spaces at baseline predicted worse BCVA at 12 months. Representative cases with or without a high area ratio and large number of cystoid spaces at baseline are shown with their outcome data at 12 months in Figure 3.

### 3.4. Association of the Area Ratio and Number of Cystoid Spaces at Baseline with Outer Retinal Morphology at 12 Months

BCVA at 12 months was positively correlated with the extents of ELM disruption (*R* = 0.577, *p* < 0.001, Figure 4a) and EZ disruption (*R* = 0.589, *p* < 0.001, Figure 4b) at 12 months. Moreover, the extents of both ELM and EZ disruptions at 12 months were positively correlated with the area ratio (ELM disruption, *R* = 0.377, *p* = 0.009, Figure 4c; EZ disruption, *R* = 0. 468, *p* = 0.001, Figure 4d), and number (ELM disruption, *R* = 0.485, *p* < 0.001, Figure 4e; EZ disruption, *R* = 0. 483, *p* < 0.001, Figure 4f) of the intraretinal cystoid spaces at baseline.

Multiple linear regression models showed that the area ratio and number of cystoid spaces at baseline were positively correlated with ELM (*R* = 0.321, *p* = 0.044 for the area ratio; *R* = 0.406, *p* = 0.007 for number) and EZ (*R* = 0.353, *p* = 0.025 for the area ratio; *R* = 0.372, *p* = 0.014 for number) disruptions at 12 months after adjusting for age and initial BCVA (Table 3). This indicated that a high area ratio and large number of intraretinal cystoid spaces at baseline resulted in greater extents of ELM and EZ disruptions at 12 months. 

### 3.5. Association of the Area Ratio and Number of Cystoid Spaces at Baseline and CRT at the Time of DME Resolution

Next, we analyzed the CRT of 37 eyes whose cystoid spaces and DME had resolved at least once during the 12-month follow up; the other nine eyes did not achieve complete resolutions of DME during these 12 months. The mean CRT at the time of DME resolution was 204 ± 8 μm, and the CRT at that time point was correlated with BCVA at 12 months (*R* = −0.369; 95% CI, −0.004 to −0.0003; *p* = 0.024; Figure 5a). This indicated that a thinner CRT at the time of DME resolution resulted in worse BCVA at 12 months. 

Finally, we found that CRT at the time of cystoid space resolution was negatively correlated with the area ratio (*R* = −0.707; 95% CI, −0.839 to −0.496; *p* < 0.001; Figure 5b) and number (*R* = −0.561; 95% CI, −0.749 to −0.290; *p* < 0.001; Figure 5c) of intraretinal cystoid spaces at baseline. 

A representative case with a high area ratio and large number of cystoid spaces, with a thin CRT at the time of DME resolution, is shown in Figure 6. In this eye, the extents of ELM and EZ disruptions were increased at the time of DME resolution.

## 4. Discussion

Here, we examined initial OCT images to determine biomarkers that can be easily applied in daily clinical practice to predict the visual outcome after IVR monotherapy. Overall, the patients with DME were successfully treated with IVR monotherapy in terms of both visual function and retinal morphology (Figure 1). The area ratio and number of intraretinal cystoid spaces at baseline predicted BCVA at 12 months (Table 2). The OCT findings for the cystoid spaces were also positively correlated with the extents of EZ and ELM disruptions at 12 months (Figure 2) and negatively correlated with CRT at the time of DME resolution (Figure 3). The extents of disruptions and the CRT were related to BCVA at 12 months. 

One of the main molecular mechanisms underlying DME is BRB disruption by inflammatory cytokines such as VEGF [21]. Ranibizumab was the first anti-VEGF agents approved by the US Food and Drug Administration for the treatment of DME, and several studies have proven the safety and efficacy of ranibizumab for the treatment of DME [13,22,23,24,25,26]. In the current study, IVR treatment significantly improved BVCA and OCT findings on average; both CRT and the area ratio and number of intraretinal cystoid spaces—both of which represent DME—and DRIL and ELM and EZ disruptions, all three of which correspond to neural retinal damage, started to improve by 3 months after initial injection, and the improvement lasted for or became significant by 12 months.

OCT images are essential for managing the treatment of DME, and various predictive factors obtained from OCT images have been reported for anti-VEGF therapy. CRT is a major clinical factor for macular diseases, and a previous report showed that the correlation between CRT and BCVA after treatment was modest in DME [27]. In the current study, the initial CRT was not associated with the visual outcome. Disruptions of the ELM and EZ, which are parameters representing photoreceptor misalignment, have been shown to correlate with worse visual outcomes in the natural course of and after vitrectomy in DME [28,29]; however, in the current study with anti-VEGF therapy, disruptions of the ELM and EZ at baseline were not correlated with BCVA at 12 months according to the multivariate analyses after adjusting for age and initial BCVA, possibly because the patients with disruptions of the ELM and EZ, and photoreceptor damage, may have already had worse BCVA at baseline, rendering the initial BCVA a confounder. Santos et al. reported that the comparison of lower than normal optical reflectivity—which corresponds to blood–retinal barrier impairment and the resulting abnormal accumulation of intra- and sub-retinal fluid—before and 1-week after the initial IVR predicted the visual outcome [30]. This method may predict the responsiveness of the exudative changes to IVR, and help in determining the following IVR treatments, although it may not be used for predicting the outcome before initial treatment or for evaluating retinal neural changes that affect visual outcomes.

By contrast, the area ratio and number of intraretinal cystoid spaces were new and significant predictive factors for the visual outcome after IVR treatment for DME irrespective of the patients’ age and initial BCVA. Therefore, when the cystoid spaces are growing and/or increasing, it may be advisable to start treatment soon, even if BCVA is acceptable and photoreceptor damage is not evident in OCT images, although further studies are warranted. It is an advantage that the intraretinal cystoid spaces on OCT images are easily evaluated visually during daily clinical practice.

Longer extents of ELM and EZ disruptions at 12 months were related to worse BCVA at 12 months. Given that the ELM and EZ disruptions reflect photoreceptor disorganization, it is natural that they would be related to to BCVA at the same time point. More importantly, a greater area ratio and number of cystoid spaces were correlated with longer extents of ELM and EZ disruptions at 12 months, suggesting that the severity of cystoid spaces may have afterward influenced the ELM and EZ conditions. In addition, eyes with a greater area ratio and number of cystoid spaces at baseline showed a very thin retina at the time of cystoid space and DME remission; a thin CRT, such as that under 200 μm, most likely indicated neural atrophy rather than a healthy retina after DME remission as previously reported. [31] Karst reported that the risk factors for atrophy with a CRT below 200 μm were higher CRT, worse BCVA, and a center involving DME at first DME presentation [32]. The height of the intraretinal cystoid space was also reported to predict the visual outcome by Gerendas et al. [33]. We showed that a greater area ratio and number, thus a special distribution of intraretinal cystoid spaces at baseline, may be biomarkers for future retinal degeneration due to DME. In other words, although the BCVA had not substantially declined and the initial ELM and EZ disruptions were not severe, severe intraretinal cystoid spaces at baseline could be associated with the onset of photoreceptor degeneration due to DME, resulting in a worse visual outcome despite treatment.

Cystoid spaces developed as a result of extracellular fluid, i.e., edema, mainly spread to the inner retinal layers [34]; however, they were related to photoreceptor findings and BCVA after treatment. This is consistent with previously reported findings that the non-perfusion area in the inner retinal layer corresponds to photoreceptor loss in branch retinal vein occlusion [35]. This could be due to the fact that the outer plexiform layer, where synapses of photoreceptor cells to the secondary neurons spread, is affected by DME in the inner retinal layer [36], which can cause subsequent photoreceptor loss [37,38,39]. 

Inner retinal changes may also affect Müller glial cells whose cell bodies are in the inner retinal layer. In particular, the human and macaque foveolas contain specialized Müller glial cells, called foveal Müller cells [40]. They provide mechanical stability and barrier function [41] and act neuroprotectively by maintaining the homeostasis of the microenvironment and by secreting neurotrophic factors [42,43,44]. A high area ratio and large number of cystoid spaces due to increased extracellular fluid and DME may affect a greater number of foveal Müller cells, which may accelerate the degeneration of surrounding neurons including photoreceptor cells.

The hypothesis that the presence of intraretinal cystoid spaces is related to the progression of photoreceptor degeneration to ultimately inflict permanent ELM and EZ disruptions and a thin retina despite treatment merits further investigation.

The limitation of this study was the relatively small sample size and the retrospective design. Moreover, multiple physicians managed the patients, and treatment requirements were determined by each physician and not by the reading center. If the patients refused additional injection although it was recommended, treatment was not performed at that time point; however, there were few such patients. The range of initial BCVA was relatively wide, and eyes with good and bad BCVA at baseline were involved; however, the significant associations were found after adjusting for age and initial BCVA. In addition, because continued treatments would be ideal to attain the best possible outcome [45], more aggressive treatment, such as a monthly regimen rather than a pro re nata regimen, could have possibly led to a better outcome. However, the current study reflects real-life world clinical conditions. The calculation of the area and number of intraretinal cystoid spaces was performed manually and not automated; [46] however, these parameters can be determined from a glance at the OCT images during daily clinical practice.

In conclusion, the area ratio and number of intraretinal cystoid spaces were predictors for the functional and morphological prognoses of IVR monotherapy for DME. A greater area ratio and number of cystoid spaces may reflect greater retinal neural vulnerability at baseline, and these may be biomarkers for the progression of retinal degeneration as shown by the ELM and EZ disruptions and/or thin retina on the OCT images despite treatment. Although further studies are needed to validate the present findings, these factors could help to predict treatment efficacy and obtain proper informed consent before treatment.

## Figures and Tables

**Figure 1 jcm-09-01391-f001:**
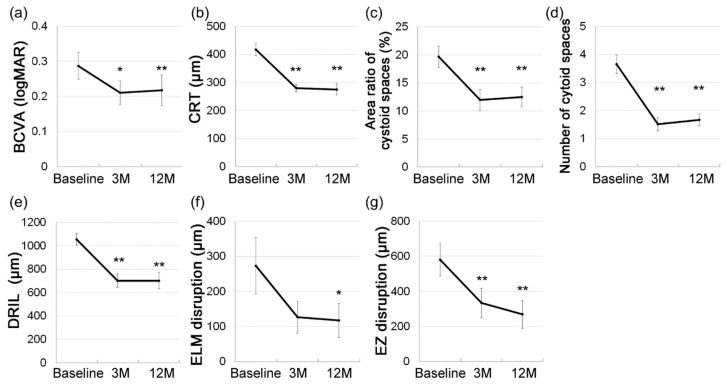
Treatment outcome of intravitreal ranibizumab monotherapy for diabetic macular edema. Repeated measure ANOVA was performed to compare the values of best-corrected visual acuity (BCVA, **a**), central retinal thickness (CRT, **b**), area ratio (**c**), number (**d**) of intraretinal cystoid spaces, extents of disorganization of the retinal inner layers (DRIL, **e**), and external limiting membrane (ELM, **f**) and ellipsoid zone (EZ, **g**) disruptions at 3 and 12 months with those at baseline. Values significantly decreased 3 and 12 months after initial treatment compared with baseline, except for ELM disruption, which showed a decrease at 3 months. Data are shown as mean ± standard error. * *p* < 0.05, ** *p* < 0.01.

**Figure 2 jcm-09-01391-f002:**
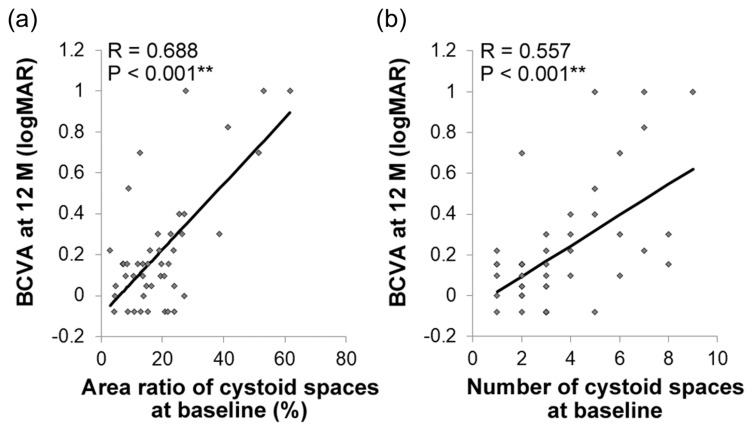
Association of the area ratio and number of intraretinal cystoid spaces at baseline with best-corrected visual acuity (BCVA) 12 months after intravitreal ranibizumab monotherapy. Pearson correlation coefficient. The area ratio (**a**) and number (**b**) of intraretinal cystoid spaces at baseline were significantly correlated with BCVA at 12 months. ** *p* < 0.01.

**Figure 3 jcm-09-01391-f003:**
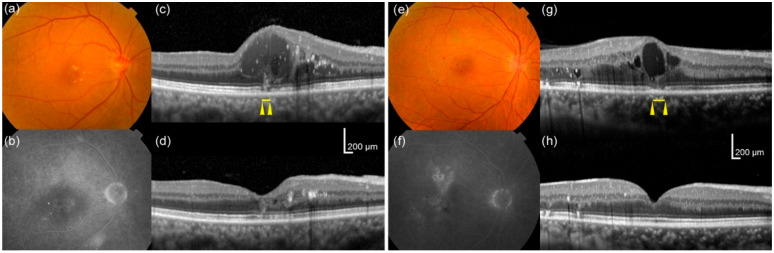
Representative cases with or without a high area ratio and large number of cystoid spaces at baseline. (**a**–**d**) A 74-year-old man whose eye had a high area ratio (38.6%) and a large number of intraretinal cystoid spaces at baseline exhibited a poor visual outcome; the best-corrected visual acuity (BCVA) was 0.222 at baseline and 0.301 at 12 months (both in LogMAR), while the extents of ELM (0 μm) and EZ (271 μm) disruptions were relatively small at baseline. (**e**–**h**) A 64-year-old man whose eye had a low area ratio (12.9%) and a small number of intraretinal cystoid spaces at baseline exhibited a good visual outcome; BCVA was 0.301 at baseline and −0.079 at 12 months (both in LogMAR). In this eye, the extents of ELM (60 μm) and EZ (220 μm) disruptions were also relatively small at baseline. Fundus color photographs (**a**,**e**), fluorescein angiograms of late phase (**b**,**f**), and optical coherence tomography (OCT) images at baseline (**c**,**g**) and at 12 months (**d**,**h**). Arrowheads (**c**,**g**) show the extents of EZ disruption.

**Figure 4 jcm-09-01391-f004:**
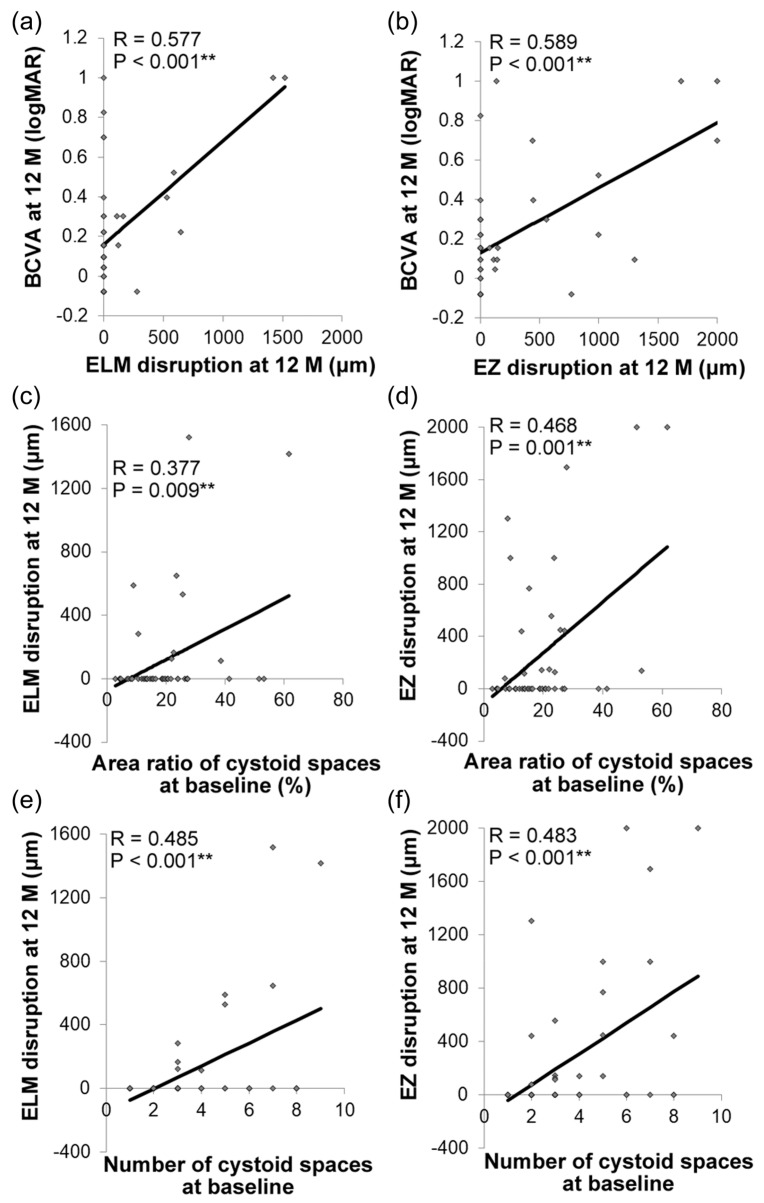
Association of the area ratio and number of intraretinal cystoid spaces at baseline with the outer retinal structures at 12 months. Pearson correlation coefficient. Best-corrected visual acuity (BCVA) at 12 months was correlated with the extents of external limiting membrane (ELM) (**a**) and ellipsoid zone (EZ) (**b**) disruptions at 12 months. The area ratio (**c**,**d**) and number (**e**,**f**) of intraretinal cystoid spaces at baseline were positively correlated with the extents of ELM and EZ disruptions at 12 months. ** *p* < 0.01.

**Figure 5 jcm-09-01391-f005:**
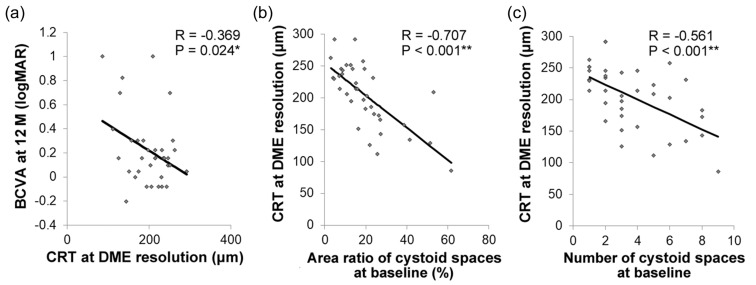
Association of the area ratio and number of intraretinal cystoid spaces at baseline with central retinal thickness (CRT) at the time of diabetic macular edema (DME) resolution. Pearson correlation coefficient. (**a**) Central retinal thickness (CRT) at the time of DME resolution was negatively correlated with best-corrected visual acuity (BCVA) at 12 months. The area ratio (**b**) and number (**c**) of intraretinal cystoid spaces at baseline were negatively correlated with CRT at the time of DME resolution. ** *p* < 0.01.

**Figure 6 jcm-09-01391-f006:**
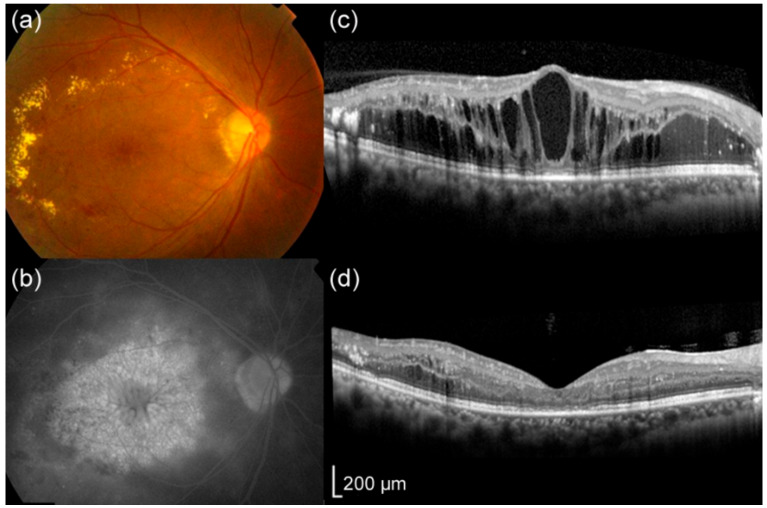
A patient whose eye had a high area ratio and a large number of intraretinal cystoid spaces at baseline who showed thin central retinal thickness (CRT) at the time of cystoid space resolution. This was a 41-year-old man with diabetic macular edema and a best-corrected visual acuity (BCVA) of 0.699 in LogMAR at baseline. A fundus color photograph (**a**), fluorescein angiogram at the late phase (**b**), and optical coherence tomography (OCT) image (**c**) at baseline. (**c**) Baseline CRT, area ratio, number of intraretinal cystoid spaces within a 2000 μm area from the foveal center, and extents of ELM and EZ disruptions were 601 μm, 41.5%, 7, 0 μm, and 503 μm, respectively. (**d**) OCT image at the time of intraretinal cystoid space resolution showing that the CRT and extents of ELM and EZ disruptions were 134 μm, 39 μm, and 1149 μm, respectively. The BCVA (LogMAR) was 0.824.

**Table 1 jcm-09-01391-t001:** Baseline characteristics.

Age (y/o, mean (range))	65.5 ± 1.6 (40–80)
Sex (male, eyes (%))	29 (63.0)
HbA1c (%, mean (range))	7.2 ± 0.1 (5.4–10.0)
Hypertension (%)	28 (60.9)
BCVA (logMAR, mean (range))	0.285 ± 0.038 (−0.079–1.000)
Fundus findings	
Mild to moderate non-proliferative DR (eyes (%))	13 (28.26)
Severe non-proliferative DR (eyes (%))	20 (43.48)
Proliferative DR (eyes (%))	13 (28.26)
OCT findings	
CRT (μm, mean (range))	418 ± 21 (167–800)
Area ratio of intraretinal cystoid spaces (%, mean (range))	19.7 ± 1.9 (2.9–61.7)
Number of intraretinal cystoid spaces (mean (range))	3.7 ± 0.3 (1–9)
Extent of DRIL (μm, mean (range))	1056 ± 51 (280–1815)
Extent of ELM disruption (μm, mean (range))	273 ± 80 (0–2000)
Extent of EZ disruption (μm, mean (range))	580 ± 95 (0–2000)
Presence of hyperreflective foci (eyes (%))	16 (34.8)
Presence of SRF (eyes (%))	9 (19.6)

Data are shown as mean ± SE. HbA1c, hemoglobin A1c; BCVA, best-corrected visual acuity; DR, diabetic retinopathy; OCT, optical coherence tomography; CRT, central retinal thickness; DRIL, disorganization of retinal inner layers; ELM, external limiting membrane; EZ, ellipsoid zone; SRF, subretinal fluid.

**Table 2 jcm-09-01391-t002:** Association between BCVA at 12 months and baseline findings.

	Crude			Multi-adjusted		
	*R*	*p*	95%CI	*R*	*p*	95%CI
Age	−0.015	0.919	−0.304 to 0.276	-	-	-
Sex (male)	0.030	0.843	−0.263 to 0.318	0.046	0.604	−0.082 to 0.139
HbA1c	−0.136	0.368	−0.410 to 0.161	−0.070	0.431	−0.079 to 0.035
Hypertension	−0.220	0.141	−0.480 to 0.075	−0.058	0.524	−0.148 to 0.075
BCVA	0.830	<0.001	0.711 to 0.903	-	-	-
Proliferative DR	0.162	0.284	−0.135 to 0.432	0.090	0.327	−0.061 to 0.179
CRT	0.311	0.036	0.023 to 0.572	0.049	0.608	−0.001 to 0.001
Area ratio of intraretinal cystoid spaces	0.688	<0.001	0.488 to 0.811	0.407	< 0.001 *	0.006 to 0.013
Number of intraretinal cystoid spaces	0.557	<0.001	0.319 to 0.730	0.241	0.012 *	0.008 to 0.059
Extent of DRIL	0.593	<0.001	0.365 to 0.753	0.168	0.119	−0.001 to 0.001
Extent of ELM disruption	0.405	0.005	0.130 to 0.622	0.029	0.778	−0.001 to 0.001
Extent of EZ disruption	0.157	0.296	−0.139 to 0.428	−0.088	0.346	−0.001 to 0.001
Presence of hyperreflective foci	−0.014	0.926	−0.303 to 0.277	0.024	0.785	−0.010 to 0.126
Presence of SRF	−0.016	0.915	−0.305 to 0.275	0.037	0.681	−0.106 to 0.160

Stepwise multiple linear regression models adjusted for age and BCVA at baseline. HbA1c, hemoglobin A1c; BCVA, best-corrected visual acuity; DR, diabetic retinopathy; CRT, central retinal thickness; DRIL, disorganization of retinal inner layers; ELM, external limiting membrane; EZ, ellipsoid zone; SRF, subretinal fluid. * *p* < 0.05.

**Table 3 jcm-09-01391-t003:** Association of area ratio and number of intraretinal cystoid spaces at baseline and outer retinal structures at 12 months.

	Area Ratio of Cystoid Space			Number of Cystoid Space		
	*R*	*p*	95%CI	*R*	*p*	95%CI
ELM disruption at 12 M	0.321	0.044 *	0.226 to 16.24	0.406	0.007 *	18.35 to 96.96
EZ disruption at 12 M	0.353	0.025 *	1.950 to 27.39	0.372	0.014 *	19.28 to 160.3

Multiple regression analyses adjusted for age and initial best-corrected visual acuity. ELM, external limiting membrane; EZ, ellipsoid zone. * *p* < 0.05.

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
