# Peer review of "The Area and Number of Intraretinal Cystoid Spaces Predict the Visual Outcome after Ranibizumab Monotherapy in Diabetic Macular Edema"

_jcm, 2020, doi:10.3390/jcm9051391_

Round 1

Reviewer 1 Report

This study is valuable in that it has demonstrated the association with long-term vision prognosis using image markers that can be intuitively judged using OCT in clinical practice.

However, there are several things to be corrected and complemented.

  1. Clinically, patients with DME often include various findings such as SRF, exudate, and retinal hemorrhage in addition to intraretinal cystoid edema, so it is thought that there will be difficulties in evaluating the correlation between visual acuity and IRE alone. And the patients with too poor or good vision can cause bias in evaluating the correlation. More detailed description of the patient's enroll criteria is required.
  2. In image analysis, if the area ratio of intraretinal cysts alone in the cases with various types of edema such as SRF is evaluated, the total amount of intraretinal cysts can be underestimated. Authors need to comment on how they handled this. In addition, the baseline characteristics require information on the type of edema observed in the OCT, and information on the absolute amount of intraretinal cysts also helps readers evaluate their clinical significance.
  3. Additionally, there is a repeated part of the sentence, so it needs to be deleted.

In the line 109, “Thirteen ~ non-proliferative DR, and 13 eyes (28.3%) had proliferative DR.” should be changed to “ Thirteen ~ non-proliferative DR.”

  1. The following sentence needs to be corrected to clarify its meaning based on the results.

In the line 255, “In other words, if the BCVA ~” should be corrected to “In other words, although the BCVA ~.”

Author Response

Point-by-Point Responses to the Reviewers’ Comments

Reviewer 1

Comments and Suggestions for Authors

This study is valuable in that it has demonstrated the association with long-term vision prognosis using image markers that can be intuitively judged using OCT in clinical practice.

However, there are several things to be corrected and complemented.

We appreciate the reviewer for understanding our work.

  1. Clinically, patients with DME often include various findings such as SRF, exudate, and retinal hemorrhage in addition to intraretinal cystoid edema, so it is thought that there will be difficulties in evaluating the correlation between visual acuity and IRE alone. And the patients with too poor or good vision can cause bias in evaluating the correlation. More detailed description of the patient's enroll criteria is required.

We agree the reviewer that the patients with DME often include various findings and we have evaluated the correlation between presence of SRF or hyperreflective foci, and visual outcome at 12 months and shown in Table 2 in the original manuscript. There were no significant correlations. There were no eyes with a pre- or intra-retinal hemorrhage greater than 1/2 DD at the foveal region in the current study. We described this point in the revised manuscript as follows;

Line 127-

There were no eyes with pre- or intra-retinal hemorrhage greater than 1/2 DD at the foveal region.

We included all the cases who were treated with intravitreal ranibizumab monotherapy, and BCVA at baseline ranged from -0.079 to 1.000 in LogMAR. In fact, eyes which had too poor vision, such as below 1.0 in LogMAR (20/200 in Snellen chart), were not involved probably because patients with too poor BCVA had given up treatment. However, the range of initial BCVA was relatively wide. This point was included in the limitation paragraph of the revised manuscript as follows;

Line 316-

The range of initial BCVA was relatively wide, and eyes with good and bad BCVA at baseline were involved; however, the significant associations were found after adjusting for age and initial BCVA.

  1. In image analysis, if the area ratio of intraretinal cysts alone in the cases with various types of edema such as SRF is evaluated, the total amount of intraretinal cysts can be underestimated. Authors need to comment on how they handled this. In addition, the baseline characteristics require information on the type of edema observed in the OCT, and information on the absolute amount of intraretinal cysts also helps readers evaluate their clinical significance.

We did not include SRF in either cystoid or retinal areas. Therefore, the exudative change related to DME could be underestimated. Nonetheless, the cystoid space area significantly affected the visual outcome. The method how we handle SRF for measurement was described in the revised manuscript as follows;

Line 93-

SRF was not involved either in the intraretinal cystoid space or retinal areas.

Types of edema and absolute amount of intraretinal cystoid space were added in the revised manuscript as follows;

Line 125-

All eyes (100%) exhibited CME, and 37 eyes (80.4%) were categorized into CME type with no SRD type, 9 eyes (19.6%) were SRD type, and no eyes were sponge-like retinal swelling type. [9]

Line 119-

The mean area ratio of the intraretinal cystoid space to the retinal area was 19.7 ± 1.9%, and the mean number of cystoid spaces was 3.7 ± 0.3; the area ratio was calculated using data of absolute amount of the intraretinal cystoid space [mean amount, 8356 ± 994 pixel2; range, 997 – 37290 pixel2] and the retinal [mean amount, 39895 ± 1442 pixel2; range, 21885 – 72565 pixel2] areas (data not shown).

  1. Additionally, there is a repeated part of the sentence, so it needs to be deleted. In the line 109, “Thirteen ~ non-proliferative DR, and 13 eyes (28.3%) had proliferative DR.” should be changed to “ Thirteen ~ non-proliferative DR.”

We graded the DR severity into 3 categories. Then there were 13 mild to moderate non-proliferative, 20 severe non-proliferative, and 13 proliferative DR eyes. So, there were no repeated parts.

  1. The following sentence needs to be corrected to clarify its meaning based on the results.In the line 255, “In other words, if the BCVA ~” should be corrected to “In other words, although the BCVA ~.”

We revised the sentence accordingly.

Reviewer 2 Report

Authors demonstrated that the amount of cystoid spaces is a novel predictor of efficacy of anti-VEGF treatment for DME and explored their association with structural changes.  The manuscript is well organized and the minor points were shown below.

1.      Abstract    The authors stated that the factors were related to retinal degeneration in DME, although the data did not support it.

2.      Methods lines 73-75  Were intraretinal or subretinal exudative changes evaluated on fundus examination or OCT?

3.      Lines 80-92  Did they assess OCT findings on sectional or three-dimensional images? I would describe the image acquisition protocols.

4.      Results  How was pan-retinal photocoagulation at baseline or during the follow-up?

5.      How did they determine the DR severity grades? International severity scale? If so, the severity should be decided according to fundus examination. Non-proliferative DR means moderate non-proliferative DR?

6.      Discussion  line 291   these factors could help to determine treatment efficacy  I would exchange ‘determine’ to ‘predict’.

Author Response

Point-by-Point Responses to the Reviewers’ Comments

Reviewer 2

Comments and Suggestions for Authors

Authors demonstrated that the amount of cystoid spaces is a novel predictor of efficacy of anti-VEGF treatment for DME and explored their association with structural changes. The manuscript is well organized and the minor points were shown below.

We appreciate the reviewer for understanding our work and providing constructive advice to improve our manuscript.

  1. Abstract   The authors stated that the factors were related to retinal degeneration in DME, although the data did not support it.

We intended to say that The area ratio and number of intraretinal cystoid spaces were related to ELM and EZ changes, and retinal atrophy, so we revised the phrase as follows;

Line 28-

The factors were related to retinal neurodegenerative changes in DME and can help obtain proper informed consent before treatment.

  1. Methods lines 73-75 Were intraretinal or subretinal exudative changes evaluated on fundus examination or OCT?

The clinicians mainly used OCT for evaluating the intraretinal or subretinal exudative changes, referring to the results of fundus examination. We added the description as follows;

Line 76-

Exudative changes were evaluated mainly by OCT images, referring to the results of fundus examinations.

  1. Lines 80-92 Did they assess OCT findings on sectional or three-dimensional images? I would describe the image acquisition protocols.

We used sectional images as described in the original manuscript. We added the protocol as follows;

Line 83-

Horizontal sections of OCT images passing through the fovea were obtained with a Heidelberg Spectralis OCT (Heidelberg Engineering, Dossenheim, Germany) instrument by OCT Section mode (30°, average of maximum 100 frames acquired using retinal tracking system). Repeat mode was applied at the time of follow-up visits to obtain OCT images at the exact same location.

  1. Results How was pan-retinal photocoagulation at baseline or during the follow-up?

Twenty-six eyes (56.5%) had undergone pan-photocoagulation before initial treatment, and no eyes had after the initial treatment up to month 12. We have already included the latter point in the original manuscript that the only treatment the patients received was IVR, we included the former point in the revised manuscript as follows;

Line 117-

Twenty-six eyes (56.5%) had past histories of pan-retinal photocoagulation.

  1. How did they determine the DR severity grades? International severity scale? If so, the severity should be decided according to fundus examination. Non-proliferative DR means moderate non-proliferative DR?

Yes, we used international severity scale. We determined the DR severity grade by using fundus photograph, and referring to the FA at baseline. Non-proliferative DR means mild to moderate non-proliferative DR. We revised the description in the text and Table 1 for accuracy.

Line 77-

Severity of DR was graded by using fundus photograph, and referring to the FA at baseline.

Line 115-

Thirteen eyes (28.26%) had mild to moderate non-proliferative DR, 20 eyes (43.48%) had severe non-proliferative DR, and 13 eyes (28.26%) had proliferative DR.

  1. Discussion line 291   these factors could help to determine treatment efficacy I would exchange ‘determine’ to ‘predict’.

Thank you for your suggestion. We revised the word accordingly.

Reviewer 3 Report

I carefully read the article entitled “The area and number of intraretinal cystoid spaces predict the visual outcome after ranibizumab monotherapy in diabetic macular edema” by Norihiro Nagai et al.

Biomarkers of visual acuity response to  anti-VEGF treatment in diabetic macular edema have been widely investigated in literature. The qualitative and quantitative parameters in foveal cystoid spaces have been correlated with anatomical responses to medical treatments. Vitreoretinal abnormalities have been identified as a predictor of lower efficacy of IVR, whereas  the ellipsoid zone of photoreceptors line or the IS/OS segment, marker of foveal photoreceptor status, are restored under anti-VEGF treatment, explaining their functional efficacy (Mori Y, et al. Restoration of foveal photoreceptors after intravitreal ranibizumab injections for diabetic macular edema. Sci Rep. 2016;6:39161). Norihiro Nagai et al. identified the area ratio and number of intraretinal cystoid spaces as significant predictive factors for the visual outcome after IVR treatment for DME. A greater area ratio and number of cystoid spaces may reflect retinal neural vulnerability at baseline, and these may be biomarkers for the progression of retinal degeneration as shown by the ELM and EZ disruptions and/or thin retina on the OCT images despite treatment. This concept doesn’t seem to hold any novelty; it appears quite obvious and an expected consequence that a severe intraretinal cystoid edema will bring to a photoreceptor degeneration in a 12 months follow-up despite treatment.Therefore a more supportive explanation should be built.It might be helpful to explain in detail how they quantify the improvement or restoring of ELM and EZ status in cases where it is shown. Images comparing eyes that underwent more or less the same number of IVR injection, with different characteristic at baseline regarding the area ratio and number of intraretinal cystoid spaces and the different outcome on ELM and EZ disruptions could be added. A comparing figure might be enclosed to fig. 5, maybe with arrows to address the most compromised areas of retinal disruption along with BCVA info; for example an eye that had a large area ratio and increased number of intraretinal cystoid spaces at baseline but showed a healthier ELM and EZ status versus an eye with smaller area ratio and less number of intraretinal cystoid spaces but with a bad ELM and EZ status, if this is the aim of their findings. Otherwise I don’t see any point in this retrospective investigation.

A recent article evaluated the effects of anti-VEGF treatment on retinal fluid in patients with diabetic macular edema through OCT leakage to quantify sites of lower than normal optical reflectivity and correlate these findings with best-corrected visual acuity response ( Santos AR et al. MEASUREMENTS OF RETINAL FLUID BY OPTICAL COHERENCE TOMOGRAPHY LEAKAGE IN DIABETIC MACULAR EDEMA: A Biomarker of Visual Acuity Response to Treatment. Retina 2019 Jan;39(1):52-60). This paper should be mentioned by Authors and they should evidence the differences and novelty of their results.

Author Response

Point-by-Point Responses to the Reviewers’ Comments

Reviewer 3

Comments and Suggestions for Authors

I carefully read the article entitled “The area and number of intraretinal cystoid spaces predict the visual outcome after ranibizumab monotherapy in diabetic macular edema” by Norihiro Nagai et al.

Biomarkers of visual acuity response to anti-VEGF treatment in diabetic macular edema have been widely investigated in literature. The qualitative and quantitative parameters in foveal cystoid spaces have been correlated with anatomical responses to medical treatments. Vitreoretinal abnormalities have been identified as a predictor of lower efficacy of IVR, whereas  the ellipsoid zone of photoreceptors line or the IS/OS segment, marker of foveal photoreceptor status, are restored under anti-VEGF treatment, explaining their functional efficacy (Mori Y, et al. Restoration of foveal photoreceptors after intravitreal ranibizumab injections for diabetic macular edema. Sci Rep. 2016;6:39161). Norihiro Nagai et al. identified the area ratio and number of intraretinal cystoid spaces as significant predictive factors for the visual outcome after IVR treatment for DME. A greater area ratio and number of cystoid spaces may reflect retinal neural vulnerability at baseline, and these may be biomarkers for the progression of retinal degeneration as shown by the ELM and EZ disruptions and/or thin retina on the OCT images despite treatment. This concept doesn’t seem to hold any novelty; it appears quite obvious and an expected consequence that a severe intraretinal cystoid edema will bring to a photoreceptor degeneration in a 12 months follow-up despite treatment. Therefore a more supportive explanation should be built. It might be helpful to explain in detail how they quantify the improvement or restoring of ELM and EZ status in cases where it is shown.

We quantified the ELM and EZ status by measuring the extent of ELM and EZ disruptions in the horizontal OCT images, and the mean value at each time point was shown in Figure 1. Measurement was performed by referring to the scale bars and the software embedded in the SD-OCT system, and repeated measure ANOVA was performed to compare the values, as already described in the original manuscript. If the disruptions were not observed at the time point, the value was set to 0. We added this point as follows;

Line 96-

If there were no DRIL, or disruptions of EZ, and ELM, the values were set to 0.

Images comparing eyes that underwent more or less the same number of IVR injection, with different characteristic at baseline regarding the area ratio and number of intraretinal cystoid spaces and the different outcome on ELM and EZ disruptions could be added. A comparing figure might be enclosed to fig. 5, maybe with arrows to address the most compromised areas of retinal disruption along with BCVA info; for example an eye that had a large area ratio and increased number of intraretinal cystoid spaces at baseline but showed a healthier ELM and EZ status versus an eye with smaller area ratio and less number of intraretinal cystoid spaces but with a bad ELM and EZ status, if this is the aim of their findings. Otherwise I don’t see any point in this retrospective investigation.

Thank you very much for your comment. Original Figure 5 (revised to Figure 6 in the revised manuscript) was to demonstrate that the eyes which had a high area ratio and a large number of intraretinal cystoid spaces at baseline showed very thin central retinal thickness (CRT) and atrophic retina at the time of cystoid space resolution, and not for comparing outcome of ELM and EZ disruptions. However, as you pointed, the disruptions were increased in this eye. We added a description in the revised text, and extents of ELM and EZ disruptions in the figure legend as follows;

Line 221-

In this eye, the extents of ELM and EZ disruptions were increased at the time of DME resolution.

Line 234-

(c) Baseline CRT, area ratio, and number of intraretinal cystoid spaces within a 2000 μm area from the foveal center, extents of ELM and EZ disruptions were 601 μm, 41.5%, 7, 0 μm, and 503 μm, respectively. (d) OCT image at the time of intraretinal cystoid space resolution showing that the CRT, and extents of ELM and EZ disruptions was were 134 μm, 39 μm, and 1149 μm, respectively.

One of our aims was that even if the ELM and EZ disruptions were small at baseline, the eyes which had greater cystoid space area and/or number will have worse visual prognosis. So, we presented additional cases of the eyes which had less ELM and EZ disruptions and a high area ratio and large number (revised Figure 3a-d), and less ELM and EZ disruptions and a low area ratio and small number (revised Figure 3e-h) of cystoid space at baseline, with their BCVA at baseline and at 12 months.

Figure 3. Representative cases with or without a high area ratio and large number of cystoid spaces at baseline. (a-d) A 74-year-old man whose eye had a high area ratio (38.6%) and a large number of intraretinal cystoid spaces at baseline exhibited poor visual outcome; the best-corrected visual acuity (BCVA) was 0.222 at baseline and 0.301 at 12 months (both in LogMAR), while the extents of ELM (0 μm) and EZ (271 μm) disruptions were relatively small at baseline. (e-h) A 64-year-old man whose eye had a low area ratio (12.9%) and a small number of intraretinal cystoid spaces at baseline exhibited good visual outcome; BCVA was 0.301 at baseline and -0.079 at 12 months (both in LogMAR). In this eye, the extents of ELM (60 μm) and EZ (220 μm) disruptions were also relatively small at baseline. Fundus color photographs (a, e), fluorescein angiograms of late phase (b, f), and optical coherence tomography (OCT) images at baseline (c, g) and at 12 months (d, h). Arrowheads (c, g) show extent of EZ disruption.

A recent article evaluated the effects of anti-VEGF treatment on retinal fluid in patients with diabetic macular edema through OCT leakage to quantify sites of lower than normal optical reflectivity and correlate these findings with best-corrected visual acuity response ( Santos AR et al. MEASUREMENTS OF RETINAL FLUID BY OPTICAL COHERENCE TOMOGRAPHY LEAKAGE IN DIABETIC MACULAR EDEMA: A Biomarker of Visual Acuity Response to Treatment. Retina 2019 Jan;39(1):52-60). This paper should be mentioned by Authors and they should evidence the differences and novelty of their results.

Thank you very much. We added a reference of the abovementioned paper in the revised manuscript as follows;

Line 267-

Santos et al. reported that comparison of lower than normal optical reflectivity, which corresponds to blood–retinal barrier impairment and resulting abnormal accumulation of intra- and sub- retinal fluid, before and 1-week after the initial IVR predicted visual outcome. [30] This method may predict the responsiveness of the exudative changes to IVR, and help determining the following IVR treatments, although it may not be used for predicting the outcome before initial treatment, and for evaluating retinal neural changes that affect visual outcome.

Round 2

Reviewer 3 Report

Authors fully respond to reviewer's suggestion